# LEARNING STRIDES IN CONVOLUTIONAL NEURAL NETWORKS

**Rachid Riad**[*1]**, Olivier Teboul**[2]**, David Grangier**[2] **& Neil Zeghidour**[2]
[1]ENS, INRIA, INSERM, UPEC, PSL Research University
[2]Google Research
`rachid.riad@ens.fr,{teboul, grangier, neilz}@google.com`

## ABSTRACT

Convolutional neural networks typically contain several downsampling operators, such as strided convolutions or pooling layers, that progressively reduce the resolution of intermediate representations. This provides some shift-invariance while reducing the computational complexity of the whole architecture. A critical hyperparameter of such layers is their stride: the integer factor of downsampling. As strides are not differentiable, finding the best configuration either requires cross-validation or discrete optimization (e.g. architecture search), which rapidly become prohibitive as the search space grows exponentially with the number of downsampling layers. Hence, exploring this search space by gradient descent would allow finding better configurations at a lower computational cost. This work introduces DiffStride, the first downsampling layer with learnable strides. Our layer learns the size of a cropping mask in the Fourier domain, that effectively performs resizing in a differentiable way. Experiments on audio and image classification show the generality and effectiveness of our solution: we use DiffStride as a drop-in replacement to standard downsampling layers and outperform them. In particular, we show that introducing our layer into a ResNet-18 architecture allows keeping consistent high performance on CIFAR10, CIFAR100 and ImageNet even when training starts from poor random stride configurations. Moreover, formulating strides as learnable variables allows us to introduce a regularization term that controls the computational complexity of the architecture. We show how this regularization allows trading off accuracy for efficiency on ImageNet.

## 1 INTRODUCTION

Convolutional neural networks (CNNs) (Fukushima, 1980; LeCun et al., 1989) have been the most widely used neural architecture across a wide range of tasks, including image classification (Krizhevsky et al., 2012; He et al., 2016a; Huang et al., 2017; Bello et al., 2021), audio pattern recognition (Kong et al., 2020), text classification (Conneau et al., 2017), machine translation (Gehring et al., 2017) and speech recognition (Amodei et al., 2016; Sercu et al., 2016; Zeghidour et al., 2018). Convolution layers, which are the building block of CNNs, project input features to a higher-level representation while preserving their resolution. When composed with non-linearities and normalization layers, this allows for learning rich mappings at a constant resolution, e.g. autogressive image synthesis (van den Oord et al., 2016). However, many tasks infer high-level low-resolution information (identity of a speaker (Muckenhirn et al., 2018), presence of a face (Chopra et al., 2005)) by integrating over low-level, high-resolution measurements (waveform, pixels). This integration requires extracting the right features, discarding irrelevant information over several downsampling steps. To that end, pooling layers and strided convolutions aggressively reduce the resolution of their inputs, providing several benefits. First, they act as a bottleneck that forces features to focus on information relevant to the task at hand. Second, pooling layers such as low-pass filters (Zhang, 2019) improve shift-invariance. Third, a reduced resolution implies a reduced number of floating-point operations and a higher receptive field in the subsequent layers.

---

[*]This work was conducted while interning at Google.

Pooling layers can usually be decomposed into two basic steps: (1) computing local statistics densely over the whole input (2) sub-sampling these statistics by an integer striding factor. Past work has mostly focused on improving (1), by proposing better alternatives to max and average pooling that avoid aliasing (Zhang, 2019; Fonseca et al., 2021), preserve the important local details (Saeedan et al., 2018), or adapt to the training data distribution (Gulcehre et al., 2014; Lee et al., 2016). Observing that integer strides reduce resolution too quickly (e.g. a $(2, 2)$ striding reduces the output size by 75%), Graham (2014) proposed fractional max-pooling, that allows for fractional (i.e. rational) strides, allowing for integration of more downsampling layers into a network. Similarly, Rippel et al. (2015) introduce spectral pooling which, by cropping its inputs in the Fourier domain, performs downsampling with fractional strides while emphasizing lower frequencies.

While fractional strides give more flexibility in designing downsampling layers, they increase the size of an already gigantic search space. Indeed, as strides are hyperparameters, finding the best combination requires cross-validation or architecture search (Zoph & Le, 2017; Baker et al., 2017; Tan et al., 2019), which rapidly become infeasible as the number of configurations grows exponentially with the number of downsampling layers. This led Zoph & Le (2017) not to search for strides in most of their experiments. Talebi & Milanfar (2021) and Jin et al. (2021) proposed a neural network that learns a resizing function for natural images, but the scaling factor (i.e. the stride) still required cross-validation. Thus, the nature of strides as hyperparameters — rather than trainable parameters — hinders the discovery of convolutional architectures and learning strides by backpropagation would unlock a virtually infinite search space.

In this work, we introduce DiffStride, the first downsampling layer that learns its strides jointly with the rest of the network. Inspired by Rippel et al. (2015), DiffStride casts downsampling in the spatial domain as cropping in the frequency domain. However, and unlike Rippel et al. (2015), rather than cropping with a fixed bounding box controlled by a striding hyperparameter, DiffStride learns the size of its cropping box by backpropagation. To do so, we propose a 2D version of an attention window with learnable size proposed by Sukhbaatar et al. (2019) for language modeling. On five audio classification tasks, using DiffStride as a drop-in replacement to strided convolutions improves performance overall while providing interpretability on the optimal per-task receptive field. By integrating DiffStride into a ResNet-18 (He et al., 2016a), we show on CIFAR (Krizhevsky et al., 2009) and ImageNet (Deng et al., 2009) that even when initializing strides randomly, our model converges to the best performance obtained with the properly cross-validated strides of He et al. (2016a). Moreover, casting strides as learnable parameters allows us to propose a regularization that directly minimizes computation and memory usage. We release our implementation of DiffStride[1].

## 2 METHODS

We first provide background on spatial and spectral pooling, and propose DiffStride for learning strides of downsampling layers. We focus on 2D CNNs since they are generic enough to be used for image (LeCun et al., 1989; Krizhevsky et al., 2012; He et al., 2016a) and audio (Amodei et al., 2016; Kong et al., 2020) processing (taking time-frequency representations as inputs). However, these methods are equally applicable to the 1D (e.g. time-series) and 3D (e.g. video) cases.

### 2.1 NOTATIONS

Let $x \in \mathbb{R}^{H \times W}$, its Discrete Fourier Transform (DFT) $y = \mathcal{F}(x) \in \mathbb{C}^{H \times W}$ is obtained through the decomposition on a fixed set of basis filters (Lyons, 2004):

$$\mathcal{F}(x)_{mn} = \frac{1}{\sqrt{HW}} \sum_{h=0}^{H-1} \sum_{w=0}^{W-1} x_{hw} e^{-2\pi i \left( \frac{mh}{H} + \frac{nw}{W} \right)}, \forall m \in \{0, \ldots, H-1\}, \forall n \in \{0, \ldots, W-1\}.$$

(1)

The DFT transformation is linear and its inverse is given by its conjugate $\mathcal{F}(.)^{-1} = \mathcal{F}(.)^*$. The Fourier transform of a real-valued signal $x \in \mathbb{R}^{H \times W}$ being *conjugate symmetric* (Hermitian-symmetry), we can reconstruct $x$ from the positive half frequencies for the width dimension and omit the negative frequencies ($y_{mn} = y^*_{(H-m)\bmod H,(W-n)\bmod W}$). In addition, the DFT and its inverse are differentiable with regard to their inputs and the derivative of the DFT (resp. inverse DFT)

---

[1]https://github.com/google-research/diffstride

is its conjugate linear operator, i.e. the inverse DFT (resp. DFT). More formally, if we consider $\mathcal{L} : \mathbb{C}^{H \times W} \longrightarrow \mathbb{R}$ as a loss taking as input the Fourier representation $y$, we can compute the gradient of $\mathcal{L}$ with regard to $x$, by using the inverse DFT:

$$x \in \mathbb{R}^{H \times W}, y = \mathcal{F}(x), \frac{\partial \mathcal{L}}{\partial x} = \mathcal{F}^*(\frac{\partial L}{\partial y}) = \mathcal{F}^{-1}(\frac{\partial L}{\partial y}). \tag{2}$$

We denote by $L$ the total number of convolution layers in a CNN architecture and each layer is indexed by $l$. The $\circ$ symbol represents the element-wise product between two tensors, $\lfloor . \rfloor$ is the floor operation and $\otimes$ the outer product between two vectors. $S$ represents the stride parameters, and $sg$ is the stop gradient operator (Bengio et al., 2013; Yin et al., 2019), defined has the identity function during forward pass and with zero partial derivatives.

## 2.2 DOWNSAMPLING IN CONVOLUTIONAL NEURAL NETWORKS

A basic mechanism for downsampling representations in a CNN is strided convolutions which jointly convolve inputs and finite impulse response filters and downsample the output. Alternatively, one can disentangle both operations by first applying a non-strided convolution followed by a pooling operation that computes local statistics (e.g. using an average, max (Boureau et al., 2010)) before downsampling. In both settings, downsampling does not benefit from the global structure of its inputs and can discard important information (Hinton, 2014; Saeedan et al., 2018). Moreover, and as observed by Graham (2014), the integer nature of strides only allows for drastic reductions in resolution: a 2D-convolution with strides $S = (2, 2)$ reduces the dimension of its inputs by $75\%$. Furthermore, stride configurations are cumbersome to explore as the number of stride combinations grows exponentially with the number of downsampling layers. This means that cross-validation can only explore a limited subset of the stride hyperparameter configurations. This limitation is likely to translate into lower performance, as Section 3.2 shows that an inappropriate choice of strides for a ResNet-18 architecture can account for a drop of $> 18\%$ in accuracy on CIFAR-100.

## 2.3 SPECTRAL POOLING

Energy of natural signals is typically not uniformly distributed in the frequency domain, with signals such as sounds (Singh & Theunissen, 2003), images (Ruderman, 1994) and surfaces (Kuroki et al., 2018) concentrating most of the information in the lower frequencies. Rippel et al. (2015) build on this observation to introduce *spectral pooling* which alleviates the loss of information of spatial pooling, while enabling fractional downsizing factors. Spectral pooling also preserves low frequencies without aliasing, a known weakness of spatial/temporal convnets (Zhang, 2019; Ribeiro & Schön, 2021).

We consider an input $x \in \mathbb{R}^{H \times W}$ and strides $S = (S_h, S_w) \in [1, H) \times [1, W)$. First, the DFT is computed $y = \mathcal{F}(x) \in \mathbb{C}^{H \times W}$ and for simplicity we assume that the center of this matrix is the DC component — the zero frequency. Then, a bounding box of size $\lfloor \frac{H}{S_h} \rfloor \times \lfloor \frac{W}{S_w} \rfloor$ crops this matrix around its center to produce $\tilde{y} \in \mathbb{C}^{\lfloor \frac{H}{S_h} \rfloor \times \lfloor \frac{W}{S_w} \rfloor}$. Finally, this output is brought back to the spatial domain with an inverse DFT: $\tilde{x} = \mathcal{F}^{-1}(\tilde{y}) \in \mathbb{R}^{\lfloor \frac{H}{S_h} \rfloor \times \lfloor \frac{W}{S_w} \rfloor}$. In practice, $x$ is typically a multi-channel input (i.e. $x \in \mathbb{R}^{H \times W \times C}$) and the same cropping is applied to all channels. Moreover, since $x$ is real-valued and thanks to Hermitian symmetry (see Section 2.1 for more details), only the positive half of the DFT coefficients are computed, which allows saving computation and memory while ensuring that the output $\tilde{x}$ remains real-valued.

Unlike spatial pooling that requires integer strides, spectral pooling only requires integer output dimensions, which allows for much more fine-grained downsizing. Moreover, spectral pooling acts as a low-pass filter over the entire input, only keeping the lower frequencies i.e. the most relevant information in general and avoiding aliasing (Zhang, 2019). However, and similarly to spatial pooling, spectral pooling is differentiable with respect to its inputs but not with respect to its strides. Thus, one still needs to provide $S$ as hyperparameters for each downsampling layer. In this case, the search space is even bigger than with spatial pooling since strides are not constrained to integer values anymore.

## 2.4 DIFFSTRIDE

To address the difficulty of searching stride parameters, we propose DiffStride, a novel downsampling layer that allows spectral pooling to learn its strides through backpropagation. To downsample

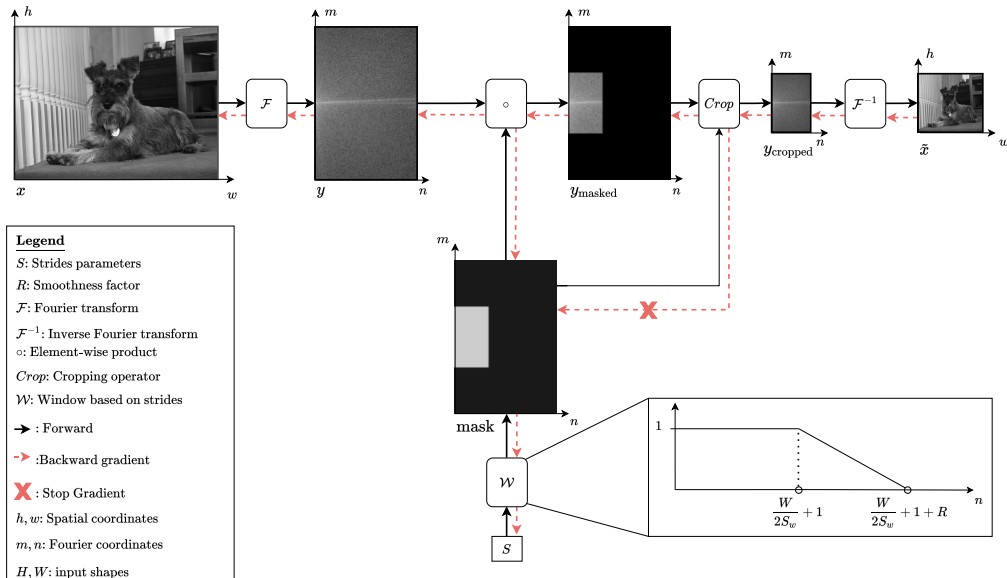

Figure 1: DiffStride forward and backward pass, using a single-channel image. We only compute the positive half of DFT coefficients along the horizontal axis due to conjugate symmetry. The zoomed frame shows the horizontal mask $\mathrm{mask}^w_{(S_w,W,R)}(n)$. Here $S = (S_h, S_w) = (2.6, 3.1)$.

$x \in \mathbb{R}^{H \times W}$, DiffStride performs cropping in the Fourier domain similarly to spectral pooling. However, instead of using a fixed bounding box, DiffStride learns the box size via backpropagation. The learnable box $\mathcal{W}$ is parametrized by the shape of the input, a smoothness factor $R$ and the strides. We design this mask $\mathcal{W}$ as the outer product between two differentiable 1D masking functions (depicted in the lower right corner of Figure 1), one along the horizontal axis and one along the vertical axis. These 1D masks are directly derived from the adaptive attention span introduced by Sukhbaatar et al. (2019) to learn the attention span of self-attention models for natural language processing. Exploiting the conjugate symmetry of the coefficients, we only consider positive frequencies along the horizontal axis, while we mirror the vertical mask around frequency zero. Therefore, the two masks are defined as follows:

$$\mathrm{mask}^h_{(S_h,H,R)}(m) = \min\left[\max\left[\frac{1}{R}(R + \frac{H}{2S_h} - |\frac{H}{2} - m|), 0\right], 1\right], m \in [0, H] \tag{3}$$

$$\mathrm{mask}^w_{(S_w,W,R)}(n) = \min\left[\max\left[\frac{1}{R}(R + \frac{W}{2S_w} + 1 - n), 0\right], 1\right], n \in [0, \frac{W}{2} + 1] \tag{4}$$

where $S = (S_h, S_w)$ are the strides and $R$ an hyperparameter that controls the smoothness of the mask. We build the 2D differentiable mask $\mathcal{W}$ as the outer product between the two 1D masks:

$$\mathcal{W}(S_h, S_w, H, W, R) = \mathrm{mask}^h_{(S_h,H,R)} \otimes \mathrm{mask}^w_{(S_w,W,R)} \tag{5}$$

We use $\mathcal{W}$ in two ways: (1) we apply it to the Fourier representation of the inputs via an element-wise product, which performs low-pass filtering (2) we crop the Fourier coefficients where the mask is zero (i.e. the output has dimensions $\lfloor \frac{H}{S_h} + 2 \times R \rfloor \times \lfloor \frac{W}{S_w} + 2 \times R \rfloor$).

The first step is differentiable with respect to strides $S$, however the cropping operation is not. Therefore, we apply a stop gradient operator (Bengio et al., 2013) to the mask before cropping. This way, gradients can flow to the strides through the differentiable low-pass filtering operation, but not through the non-differentiable cropping. Finally, the cropped tensor is transformed back into the spatial domain using an inverse DFT. All these steps are summarized by Algorithm 1 and illustrated on a single channel image in the Figure 1.

During training we constrain strides $S = (S_h, S_w)$ to remain in $[1, H] \times [1, W]$. When $x$ is a multi-channel input (i.e. $x \in \mathbb{R}^{H \times W \times C}$), we learn the same strides $S$ for all channels to ensure uniform spatial dimensions across channels. In spatial and spectral pooling, strides are typically

---

**Algorithm 1:** DiffStride layer

---

**Inputs :** Input $x \in \mathbb{R}^{H \times W}$, strides $S = (S_h, S_w) \in [1, H] \times [1, W]$, smoothness factor $R$.

**Output:** Downsampled output $\tilde{x} \in \mathbb{R}^{\lfloor \frac{H}{S_h} + 2 \times R \rfloor \times \lfloor \frac{W}{S_w} + 2 \times R \rfloor}$

1  $y \longleftarrow \mathcal{F}(x)$                                                           ▷ Project input to the Fourier domain.
2  $\text{mask} \longleftarrow \mathcal{W}(S_h, S_w, H, W, R)$                                 ▷ Construct the mask. See Equation 5.
3  $y_{\text{masked}} \longleftarrow y \circ \text{mask}$                                      ▷ Apply the mask as a low-pass filter.
4  $y_{\text{cropped}} \longleftarrow Crop(y_{\text{masked}}, sg(\text{mask}))$ ▷ Crop the tensor with the mask after stopping gradients.
5  $\tilde{x} \longleftarrow \mathcal{F}^{-1}(y_{\text{cropped}})$                             ▷ Return to the spatial domain.

---

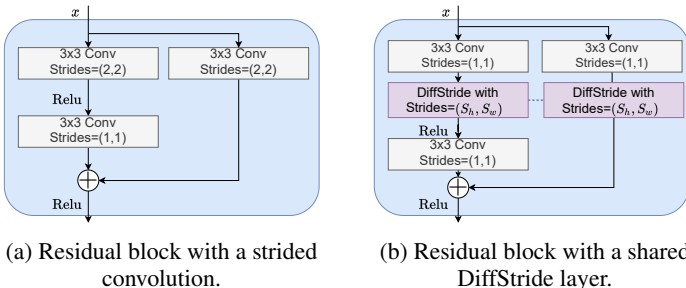

(a) Residual block with a strided convolution.     (b) Residual block with a shared DiffStride layer.

Figure 2: Comparison side by side of the shortcut blocks in classic ResNet architectures with strided convolutions, and with DiffStride that learns the strides of the block.

tied along the spatial axes (i.e. $S_w = S_h$), which we can also do in DiffStride by sharing a single learnable stride on both dimensions. However, our experiments in Section 3 show that learning specific strides for the vertical and horizontal axis is beneficial, not only when processing time-frequency representations of audio, but also — more surprisingly — when classifying natural images. Adding an hyperparameter $R$ to each downsampling layer would conflict with the goal of removing strides as hyperparameters. Thus, not only we use a single $R$ value for all layers, but we found no significant impact of this choice and all our experiments use $R = 4$. While we focus on the 2D case, using a single 1D mask allows deriving DiffStride in 1D CNNs, while performing the outer product between three 1D masks allows applying DiffStride to 3D inputs.

### 2.4.1 RESIDUAL BLOCK WITH DIFFSTRIDE

Unlike systems that only feed outputs of the $l^{th}$ layer to the $(l + 1)^{th}$ (Krizhevsky et al., 2012), ResNets (He et al., 2016a;b) introduce skip-connections that operate in parallel to the main branch. ResNets stack two types of blocks: (1) identity blocks that maintain the input channel dimension and spatial resolution and (2) shortcut blocks that increase the output channel dimension while reducing the spatial resolution with a strided convolution (see Figure 2a). We integrate DiffStride into these shortcut blocks by replacing strided convolutions by convolutions without strides followed by DiffStride. Besides, sharing DiffStride strides between the main and residual branches ensures that their respective outputs have identical spatial dimensions and can be summed (See Figure 2b).

### 2.4.2 REGULARIZING COMPUTATION AND MEMORY COST WITH DIFFSTRIDE

The number of activations in a network depends on the strides and learning these parameters gives control over the space and time complexity of an architecture in a differentiable manner. This contrasts with previous work, as measures of complexity such as the number of floating-point operations (FLOPs) are typically not differentiable with respect to the parameters of a model and searching for efficient architectures is done via high-level exploration (e.g. introducing separable convolutions (Howard et al., 2017)), architecture search (Howard et al., 2019; Tan & Le, 2019) or using continuous relaxations of complexity (Paria et al., 2020).

A standard 2D convolution with a square kernel of size $k^2$ and $C'$ output channels has a computational cost of $k^2 \times C \times C' \times H \times W$ when operating on $x \in \mathbb{R}^{H \times W \times C}$. Its memory usage— in terms of the number of activations to store— is $C' \times H \times W$. Considering a fixed number of channels and kernel size, both the computational complexity and memory usage of a convolution layer are thus linear functions of its input size $H \times W$. This illustrates our argument made in Section 1 that

downsampling does not only improve performance by discarding irrelevant information, but also reduces the complexity of the upper layers. More importantly, in the context of DiffStride the input size $H^l \times W^l$ of layer $l$ is determined as follows: $H^l \times W^l = \lfloor \frac{H^{l-1}}{S_h^{l-1}} + 2 \times R \rfloor \times \lfloor \frac{W^{l-1}}{S_w^{l-1}} + 2 \times R \rfloor$, which is differentiable with respect to the strides at the previous layer $S^{l-1}$. Furthermore, it also depends on spatial dimensions at the previous layer $H^{l-1} \times W^{l-1}$, which themselves are a function of $S^{l-2}$. By induction over layers, the total computational cost and memory usage are proportional to $\sum_{l=1}^{l=L} \prod_{i=1}^{l} \frac{1}{S_h^i \times S_w^i}$. Since in the context of DiffStride the kernel size and number of channels remain constant during training, we can directly regularize our model towards time and space efficiency by adding the following regularizer to our training loss:

$$\lambda J((S^l)_{l=1}^{l=L}) = \lambda \sum_{l=1}^{l=L} \prod_{i=1}^{l} \frac{1}{S_h^i \times S_w^i}, \tag{6}$$

where $\lambda$ is the regularization weight. In Section 3.2, we show that training on ImageNet with different values for $\lambda$ allows us to trade-off accuracy for efficiency in a smooth fashion.

## 3 EXPERIMENTS

We evaluate DiffStride on eight classification tasks, both on audio and images. For each comparison, we keep the same architecture and replace strided convolutions by convolutions with no stride followed by DiffStride. To avoid the confounding factor of downsampling in the Fourier domain, we also compare our approach to the spectral pooling of Rippel et al. (2015), which only differs from DiffStride by the fact that its strides are not learnable.

### 3.1 AUDIO CLASSIFICATION

**Experimental setup** We perform single-task and multi-task audio classification on 5 tasks: acoustic scene classification (Heittola et al., 2018), birdsong detection (Stowell et al., 2018), musical instrumental classification and pitch estimation on the NSynth dataset (Engel et al., 2017) and speech command classification (Warden, 2018). The statistics of the datasets are summarized in Table A.1. The audio sampled at $16 \text{ kHz}$ is decomposed into log-compressed mel-spectrograms with 64 channels, computed with a window of $25 \text{ ms}$ every $10 \text{ ms}$.

A 2D-CNN, based on (Tagliasacchi et al., 2019) takes these spectrograms as inputs and alternates blocks of strided convolutions along time ($(3 \times 1)$ kernel) and frequency ($(1 \times 3)$ kernel). Each strided convolution is followed by a ReLU (Glorot et al., 2011) and batch normalization (Ioffe & Szegedy, 2015). The sequence of channels dimensions is defined as $(64, 128, 256, 256, 512, 512)$ and the strides are initialized as $((2, 2), (2, 2), (1, 1), (2, 2), (1, 1), (2, 2))$ for all downsampling methods. The output of the CNN passes through a global max-pooling and feeds into a single linear classification layer for single-task, and multiple classification layers for multi-task classification. As examples vary in length, we train models on random $1 \text{ s}$ windows with ADAM (Kingma & Ba, 2015) and a learning rate of $10^{-4}$ for $1 \text{ M}$ batches, with batch size 256. Evaluation is run by splitting full sequences into $1 \text{ s}$ non-overlapping windows and averaging the logits over windows.

**Results** Table 1 summarizes the results for single-task and multi-task audio classification. In both settings, DiffStride improves over strided convolutions and spectral pooling, with strided convolutions only outperforming DiffStride for acoustic scene classification in the single task setting. Table 2 shows the strides learned by the first layer of DiffStride, which downsamples mel-spectrograms along frequency and time axes. Learning allows the strides to deviate from their initialization ($(2, 2)$) and to adapt to the task at hand. Converting strides to cut-off frequencies shows that the learned strides fall in a range showed by behavioral studies and direct neural recordings (Hullett et al., 2016; Flinker et al., 2019) to be necessary for e.g. speech intelligibility at $25 \text{ Hz}$ (Elliott & Theunissen, 2009). Moreover, DiffStride learns different strides for the time and frequency axes. Table A.7 shows the benefits of learning a per-dimension value rather than sharing strides. Another notable phenomenon is the per-task discrepancy on NSynth, with the pitch estimation requiring faster spectral modulations (as represented by a higher cutt-off frequency along the frequency axis). Finally, multi-task models do not converge to the mean of strides, but rather to a higher value that passes more frequencies not to negatively impact individual tasks.

| Setting | Single-task | | | Multi-task | | |
|---|---|---|---|---|---|---|
| Task | Strided Conv. | Spectral | DiffStride | Strided Conv. | Spectral | DiffStride |
| Acoustic scenes | **99.1** ± 0.2 | 98.6 ± 0.1 | 98.6 ± 0.2 | **97.7** ± 0.4 | **97.7** ± 0.7 | **97.7** ± 0.3 |
| Birdsong detection | 78.8 ± 0.3 | 79.7 ± 0.3 | **81.3** ± 0.1 | 77.3 ± 0.2 | 77.8 ± 0.3 | **78.6** ± 0.5 |
| Music (instrument) | 72.6 ± 0.3 | 72.9 ± 0.5 | **75.4** ± 0.0 | 69.8 ± 0.4 | 70.4 ± 0.4 | **73.0** ± 0.8 |
| Music (pitch) | 91.8 ± 0.1 | 90.1 ± 0.0 | **92.2** ± 0.1 | 89.4 ± 0.3 | 87.6 ± 0.7 | **89.9** ± 0.3 |
| Speech commands | 87.3 ± 0.1 | 88.5 ± 0.3 | **90.5** ± 0.3 | 83.5 ± 0.6 | 83.9 ± 0.4 | **86.2** ± 0.8 |
| Mean Accuracy | 85.0 ± 9.3 | 86.0 ± 9.2 | **88.3** ± 8.7 | 83.5 ± 10.0 | 83.5 ± 9.6 | **85.0** ± 8.9 |

Table 1: Test accuracy (% ± sd over 3 runs) for audio classification in the single (one model per task) and multi-task (one model for all tasks) settings.

| | Learned Strides | | Equivalent cut-off frequencies | |
|---|---|---|---|---|
| | Time | Frequency | Time (Hz) | Frequency (Cyc/Mel) |
| Acoustic scenes | 1.89 ± 0.05 | 1.99 ± 0.03 | 26.25 ± 0.63 | 0.2448 ± 0.009 |
| Birdsong detection | 1.91 ± 0.02 | 1.96 ± 0.01 | 25.83 ± 0.36 | 0.2500 ± 0.000 |
| Music (Instrument) | 1.29 ± 0.06 | 2.12 ± 0.01 | 38.33 ± 1.57 | 0.2292 ± 0.009 |
| Music (Pitch) | 1.32 ± 0.10 | 1.61 ± 0.07 | 37.50 ± 2.72 | 0.3021 ± 0.018 |
| Speech commands | 1.97 ± 0.00 | 1.95 ± 0.01 | 25.00 ± 0.00 | 0.2500 ± 0.000 |
| Multi-task model | 1.46 ± 0.01 | 1.79 ± 0.03 | 34.17 ± 0.30 | 0.2708 ± 0.0074 |

Table 2: Learned strides (% ± sd over 3 runs) of the first layer for the single and multi-task models. The sampling rate of the input spectrogram being known (10 ms), we can convert the strides to upper cut-off frequencies (i.e. the maximum frequency kept by the lowpass-filter).

## 3.2 IMAGE CLASSIFICATION

**Experimental setup** We use the ResNet-18 (He et al., 2016a) architecture, comparing the original strided convolutions (see Figure 2a) to spectral pooling and DiffStride (both as in Figure 2b). We randomly sample 6 striding configurations for the three shortcut blocks of the ResNet-18, each stride being sampled in $[1, 3]$, with $(2, 2, 2)$ being the configuration of the original ResNet of He et al. (2016a). The horizontal and vertical strides are initialized equally at start. These random configurations simulate cross-validation of stride configurations to: (1) showcase the sensitivity of the architecture to these hyperparameters, (2) test our hypothesis that DiffStride can benefit from learning its strides to recover from a poor initialization. On Imagenet, as inputs are bigger than CIFAR we also allow the first ResNet-18 identity block to learn its strides which are 1 by default.

We first benchmark the three methods on the two CIFAR datasets (Krizhevsky, 2009). CIFAR10 consists of $32 \times 32$ images labeled in 10 classes with 6000 images per class. We use the official split, with 50,000 images for training and 10,000 images for testing. CIFAR100 uses the same images as CIFAR10, but with a more detailed labelling with 100 classes. We also compare the ResNet-18 architectures on the ImageNet dataset (Deng et al., 2009), which contains 1,000 classes. The models are trained on the official training split of the Imagenet dataset (1.28M images) and we report our results on the validation set (50k images). Here, we evaluate performance in terms of top-1 and top-5 accuracy. We train on all datasets with stochastic gradient descent (SGD) (Bottou et al., 1998) with a learning rate of 0.1, a batch size of 256 and a momentum (Qian, 1999) of 0.9. On CIFAR, we train models for 400 epochs dividing the learning rate by 10 at 200 epochs and again by 10 at 300 epochs, with a weight decay of $5.10^{-3}$. For CIFAR, we apply random cropping on the input images and left-right random flipping. On ImageNet, we train with a weight decay of $1.10^{-3}$ for 90 epochs, dividing the learning rate by 10 at epochs 30, 60 and 80. We apply random cropping on the input images as in (Szegedy et al., 2015) and left-right random flipping.

**Results** We report the results on the CIFAR datasets and Imagenet in Tables 3 and 4 respectively, with the accuracy of our baseline ResNet-18 (first row, Strided Conv.) being consistent with previous work (Bianco et al., 2018). First, we observe that strides are indeed critical hyperparameters for the performance of a standard ResNet-18 on the three datasets, with the accuracy on CIFAR100 dropping from 66.8% average to 48.2% between the best and worst configurations. Remarkably, spectral pooling is much more robust to bad initializations than strided convolutions, even though

| Init. Strides | CIFAR10 | | | CIFAR100 | | |
|---|---|---|---|---|---|---|
| | Strided Conv. | Spectral | DiffStride | Strided Conv. | Spectral | DiffStride |
| $(2, 2, 2)$ | $91.4 \pm 0.2$ | $92.4 \pm 0.1$ | $\mathbf{92.5} \pm 0.1$ | $66.8 \pm 0.2$ | $\mathbf{73.7} \pm 0.1$ | $73.4 \pm 0.5$ |
| $(2, 2, 3)$ | $90.5 \pm 0.1$ | $92.2 \pm 0.2$ | $\mathbf{92.8} \pm 0.1$ | $63.4 \pm 0.5$ | $\mathbf{73.7} \pm 0.2$ | $73.5 \pm 0.0$ |
| $(1, 3, 1)$ | $90.0 \pm 0.4$ | $91.1 \pm 0.1$ | $\mathbf{92.4} \pm 0.1$ | $64.9 \pm 0.5$ | $70.3 \pm 0.3$ | $\mathbf{73.4} \pm 0.2$ |
| $(3, 1, 3)$ | $85.7 \pm 0.1$ | $90.9 \pm 0.2$ | $\mathbf{92.4} \pm 0.1$ | $55.3 \pm 0.8$ | $69.4 \pm 0.4$ | $\mathbf{73.7} \pm 0.4$ |
| $(3, 1, 2)$ | $86.4 \pm 0.1$ | $90.9 \pm 0.2$ | $\mathbf{92.3} \pm 0.1$ | $56.2 \pm 0.3$ | $69.9 \pm 0.2$ | $\mathbf{73.4} \pm 0.3$ |
| $(3, 2, 3)$ | $82.0 \pm 0.6$ | $89.2 \pm 0.2$ | $\mathbf{92.3} \pm 0.1$ | $48.2 \pm 0.2$ | $66.6 \pm 0.5$ | $\mathbf{73.6} \pm 0.4$ |
| Mean accuracy | $87.7 \pm 3.4$ | $91.1 \pm 1.1$ | $\mathbf{92.4} \pm 0.2$ | $59.1 \pm 6.7$ | $70.6 \pm 2.6$ | $\mathbf{73.5} \pm 0.3$ |

Table 3: Accuracies (% $\pm$ sd over 3 runs) on CIFAR10 and CIFAR100. First column represents strides at each shortcut block, $(2, 2, 2)$ being the configuration of (He et al., 2016a). For reference, the state-of-the-art on CIFAR10 (CIFAR100) is (Dosovitskiy et al., 2020) ((Foret et al., 2020)) with an accuracy of $99.5\%$ ($96.1\%$).

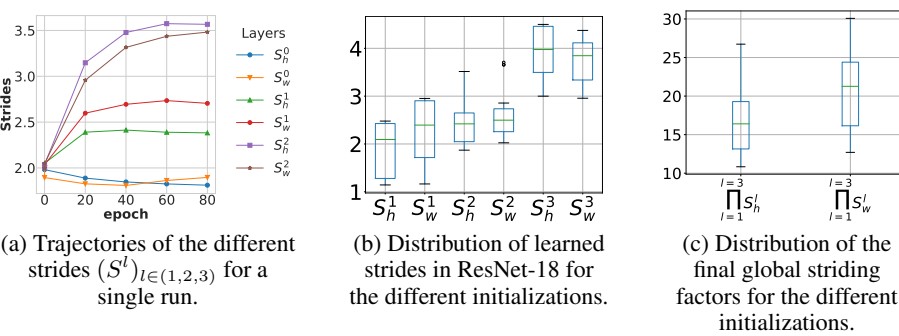

(a) Trajectories of the different strides $(S^l)_{l \in (1,2,3)}$ for a single run.

(b) Distribution of learned strides in ResNet-18 for the different initializations.

(c) Distribution of the final global striding factors for the different initializations.

Figure 3: Learning dynamics of DiffStride on the CIFAR10 dataset.

its strides are also fixed. However, DiffStride is overall much more robust to poor choices of strides, converging consistently to a high accuracy on the three datasets, with a variance over initializations that is lower by an order of magnitude. This shows that backpropagation allows DiffStride to find a better configuration during training avoiding a cross-validation which would require 6,561 experiments for testing all combinations of strides in $[1, 3]$ on Imagenet. Tables A.5 and A.6 confirm these observations on the EfficientNet-B0 (Tan & Le, 2019) architecture.

**Learning dynamics and equivalence classes** Figure 3 illustrates the learning dynamics of Diff-Stride on CIFAR10. Figure 3a plots the strides as a function of the epoch for a run with the baseline (2,2,2) configuration as initialization. The strides all deviate from their initialization while converging rapidly, with the lower layers keeping more information while higher layers downsample more drastically. Interestingly, we discover equivalence classes: despite converging to the same accuracy (as reported in Table 3) the various initializations yield very diverse strides configurations at convergence, both in terms of total striding factor (defined as the product of strides, see Figure 3c) and of repartition of downsampling factors along the architecture (see Figure 3b). We obtain similar conclusions on CIFAR100 and Imagenet (see Figures A.1 and A.2). In the non-regularized case, it could seem counter-intuitive that minimizing the training loss yields positive stride updates, i.e. dropping more information through cropping. It highlights that loss optimization is a trade-off between preserving information (no striding, no cropping) and downscaling such that the next convolution kernel accesses a wider spatial context.

**Regularizing the complexity** The existence of equivalence classes suggests that DiffStride can find more computationally efficient configurations for a same accuracy. We thus train DiffStride on ImageNet using the complexity regularizer defined in Equation 6, with $\lambda$ varying between 0.1 and 10, always initializing strides with the baseline $((1, 1), (2, 2), (2, 2), (2, 2))$. Figure 4 plots accuracy versus computational complexity (as measured by the value of the regularization term at convergence) of DiffStride. For comparison, we also plot the models with strided convolutions with the random initializations of Table 4, showing that DiffStride finds configurations with a lower computational cost for the same accuracy. Some of these are quite extreme, e.g. with $\lambda = 10$ a model converges to strides $((10.51, 32.23), (1.20, 2.68), (1.20, 2.04), (1.96, 4.53))$ for a $58.57\%$

| Init. Strides | Top-1 | | | Top-5 | | |
|---|---|---|---|---|---|---|
| | Strided Conv. | Spectral | DiffStride | Strided Conv. | Spectral | DiffStride |
| (1, 2, 2, 2) | $68.65 \pm 0.26$ | $69.01 \pm 0.19$ | **$69.66 \pm 0.06$** | $88.5 \pm 0.15$ | $88.48 \pm 0.02$ | **$89.07 \pm 0.03$** |
| (1, 1, 3, 1) | $69.79 \pm 0.15$ | **$69.88 \pm 0.05$** | $68.22 \pm 0.07$ | **$89.43 \pm 0.18$** | $89.15 \pm 0.07$ | $88.10 \pm 0.08$ |
| (1, 3, 1, 3) | $68.86 \pm 0.28$ | $68.63 \pm 0.08$ | **$69.41 \pm 0.16$** | $88.64 \pm 0.15$ | $88.42 \pm 0.01$ | **$88.98 \pm 0.04$** |
| (2, 2, 2, 3) | $63.45 \pm 0.09$ | $67.16 \pm 0.17$ | **$69.53 \pm 0.08$** | $85.09 \pm 0.04$ | $87.25 \pm 0.06$ | **$89.05 \pm 0.05$** |
| (2, 3, 1, 2) | $65.35 \pm 0.03$ | $66.35 \pm 0.24$ | **$69.42 \pm 0.06$** | $86.27 \pm 0.05$ | $86.67 \pm 0.15$ | **$88.91 \pm 0.05$** |
| (3, 3, 2, 3) | $57.11 \pm 0.11$ | $64.44 \pm 0.01$ | **$69.43 \pm 0.11$** | $80.42 \pm 0.11$ | $85.22 \pm 0.09$ | **$89.03 \pm 0.02$** |
| Mean accuracy | $65.53 \pm 4.49$ | $67.58 \pm 1.88$ | **$69.28 \pm 0.50$** | $86.39 \pm 3.15$ | $87.53 \pm 1.36$ | **$88.85 \pm 0.35$** |

Table 4: Top-1 and top-5 accuracies ($\% \pm$ sd over 3 runs) on Imagenet, $(1, 2, 2, 2)$ being the configuration of (He et al., 2016a). For reference, state-of-the-art on Imagenet is (Dai et al., 2021) with a top-1 accuracy of $90.88\%$.

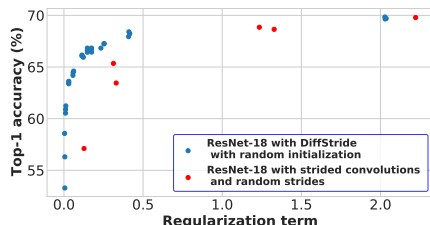

Figure 4: Top-1 accuracy (%) on the Imagenet validation set as a function of the regularization term $J\big((S^l)_{l=1}^{l=L}\big)$ as defined in equation 6, after training with $\lambda \in [0.1, 10]$.

top-1 accuracy. When training a ResNet-18 with strided convolutions using the closest integer strides (i.e. $((11, 32), (1, 3), (1, 2), (2, 5))$), the model converges to a $24.54\%$ top-1 accuracy. This suggests that performing pooling in the spectral domain is more robust to aggressive downsampling, which corroborates the remarkable advantage of spectral pooling over strided convolutions when using poor strides choices in Tables 3 and 4 despite both models having fixed strides.

**Limitations**   Pooling in the spectral domain comes at higher computational cost than strided convolutions as it requires (1) computing a non-strided convolution and (2) a DFT and its inverse (see Table A.2). This could be alleviated by computing the convolution in the Fourier domain as an element-wise multiplication and summation over channels. Further improvements could be obtained by replacing the DFT by a real-valued counterpart, such as the Hartley transform (Zhang & Ma, 2018), which would remove the need for complex-valued operations that may be poorly optimized in deep learning frameworks. We also observe no benefits of DiffStride when training DenseNets (Huang et al., 2017), see Tables A.3 and A.4. We hypothesize that this is due to the limited number of downsampling layers, which reduces the space of stride configurations to a few, equivalent ones when sampling strides in $[1; 3]$. Finally, some hardware (e.g. TPUs) require a static computation graph. As DiffStride changes the spatial dimensions of intermediate representations— and thus the computation graph— between each gradient update, we currently only train on GPUs.

## 4   CONCLUSION AND FUTURE WORK

We introduce DiffStride the first downsampling layer with learnable strides. We show on audio and image classification that DiffStride can be used as a drop-in replacement to strided convolutions, removing the need for cross-validating strides. As we observe that our method discovers multiple equally-accurate stride configurations, we introduce a regularization term to favor the most computationally advantageous. In future work, we will extend the scope of applications of DiffStride, to e.g. 1D and 3D architectures. Moreover, learning strides by backpropagation opens new avenues in designing adaptive convolutional architectures, such as multi-scale models that would learn to operate at various scales in parallel by using independent branches with separate instances of DiffStride, or by predicting strides parameters of DiffStride on a per-example basis.

## ACKNOWLEDGMENTS

Authors thank Félix de Chaumont Quitry for the useful discussions and assistance throughout this project, as well as the reviewers of ICLR 2022 for their feedback that helped improving this manuscript. Authors also thank Oren Rippel, Jasper Snoek, and Ryan P. Adams for their inspiring work on spectral pooling.

## REPRODUCIBILITY STATEMENT

We describe DiffStride in details in the text as well as with Algorithm 1 and Figure 1. We mention all relevant hyperparameters to reproduce our experiments, as well as describe audio datasets in A.1. Moreover, we release Tensorflow 2.0 code for training a Pre-Act ResNet-18 with strided convolutions, spectral pooling or DiffStride on CIFAR10 and CIFAR100, with DiffStride being implemented as a stand-alone, reusable Keras layer. This open-source code can be found at https://github.com/google-research/diffstride.

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

# A APPENDIX

Table A.1: Datasets used for audio classification. Default train/test splits are always adopted.

| Task | Name | Classes | Train examples | Test examples |
|---|---|---|---|---|
| Acoustic scenes | TUT Urban 2018 | 10 | 7,829 | 810 |
| Birdsong detection | DCASE2018 | 2 | 32,129 | 3,561 |
| Music (instrument) | Nsynth | 11 | 289,205 | 12,678 |
| Music (pitch) | Nsynth | 128 | 289,205 | 12,678 |
| Speech commands | Speech commands | 35 | 84,771 | 10,700 |

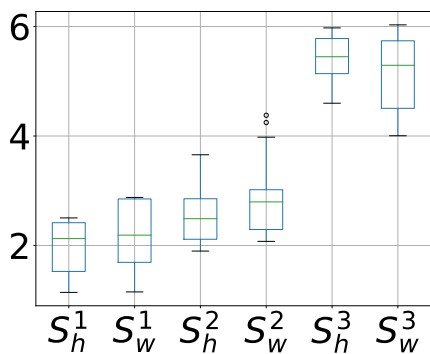
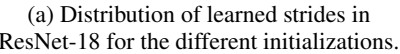
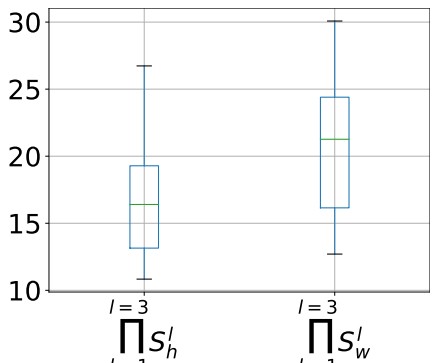

(a) Distribution of learned strides in ResNet-18 for the different initializations.

(b) Distribution of the final global striding factors for the different initializations.

Figure A.1: Learned strides by DiffStride on the CIFAR100 dataset.

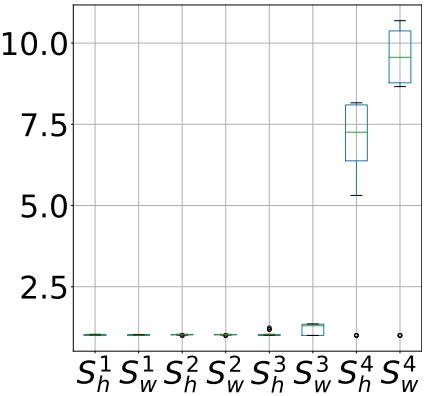
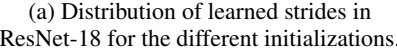
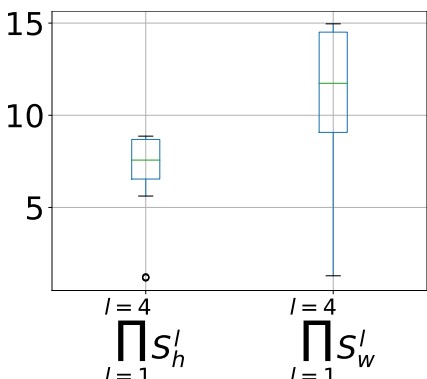

(a) Distribution of learned strides in ResNet-18 for the different initializations.

(b) Distribution of the final global striding factors for the different initializations.

Figure A.2: Learned strides by DiffStride on the Imagenet dataset.

**Analysis of strides learned on CIFAR100 and ImageNet** In Figure A.1 (Figure A.2), we show the distributions of learned strides and the global striding factor at convergence on CIFAR100 (Imagenet), starting from random stride initializations. On CIFAR100, we observe equivalence classes, i.e. model that learns various stride configurations for a same accuracy. On Imagenet, even though we also observe a significant variance of the global striding factor, models tend to downsample only in the upper layers. Striding late in the architecture comes at a higher computational cost, which furthermore justifies regularizing DiffStride to reduce complexity as shown in Section 3.2.

**Time and space complexity in practice**   While Figure 4 reports theoretical estimates of computational complexity based on stride configurations, both spectral pooling and DiffStride require computing a DFT and its inverse. Moreover, DiffStride requires accumulating gradients with respect to the strides during training. Table A.2 reports the duration and peak memory usage of the multi-task architecture described in 3.1, for a single batch. Replacing strided convolutions with spectral pooling increases the wall time by 32% due to the DFT and inverse DFT, while the peak memory usage is almost unaffected. DiffStride furthermore increases the wall time (by 43% w.r.t strided convolutions) as the backward pass is more expensive. Similarly, it almost doubles the peak memory usage. However, in inference, DiffStride does not need to compute and store gradients w.r.t. the strides, thus the time and space complexity become identical to that of spectral pooling.

|           |                  | Strided Conv. | Spectral | DiffStride |
|-----------|------------------|---------------|----------|------------|
| Training  | Time/step        | 1.0           | 1.32     | 1.43       |
|           | Peak memory (GB) | 1.0           | 1.02     | 1.98       |
| Inference | Time/step        | 1.0           | 1.32     | 1.32       |
|           | Peak memory (GB) | 1.0           | 1.02     | 1.02       |

Table A.2: Per-step time and peak memory usage of Spectral Pooling and DiffStride relative to strided convolutions, on a V100 GPU. During training, a "step" is the forward and backward pass for a single batch, while in inference it only involves a forward pass.

**DenseNet experiments on CIFAR**   We also evaluate DiffStride in DenseNet (Huang et al., 2017), especially the DenseNet-BC architecture with a depth of 121 and a growth rate of 32. The DenseNet architecture halves spatial dimensions during transition blocks. We replace the 2D average pooling in the transition blocks by spectral pooling or DiffStride. The considered architecture for DenseNet has two downsampling steps. We run a similar experiment as in 3.2 with random strides between the dense blocks on the two CIFAR datasets. We observe that initializing strides randomly does not affect the performance of the standard Densenet-BC architecture with average pooling. Consequently, DiffStride does not improve over alternatives.

| Init. Strides | Average Pooling | Spectral | DiffStride |
|---------------|-----------------|----------|------------|
| (2, 2)        | **92.3** ± 0.2  | 91.5 ± 0.2 | 91.5 ± 0.1 |
| (1, 2)        | **91.8** ± 0.2  | 90.5 ± 0.5 | 91.1 ± 0.3 |
| (1, 3)        | **92.0** ± 0.2  | 91.0 ± 0.1 | 91.6 ± 0.4 |
| (2, 3)        | 92.0 ± 0.3      | 92.1 ± 0.2 | **92.2** ± 0.1 |
| (3, 1)        | 91.6 ± 0.2      | 91.5 ± 0.4 | **91.7** ± 0.2 |
| Mean accuracy | **91.9** ± 0.3  | 91.3 ± 0.6 | 91.6 ± 0.4 |

Table A.3: Accuracies (% ± sd over 3 runs) for CIFAR10 for each downsampling method with the DenseNet-BC architecture. First column represents strides at each transition block, $(2, 2)$ being the configuration of (Huang et al., 2017).

| Init. Strides | Average Pooling | Spectral | DiffStride |
|---------------|-----------------|----------|------------|
| (2, 2)        | **69.1** ± 0.6  | 68.6 ± 0.6 | 68.2 ± 0.2 |
| (1, 2)        | 67.7 ± 0.3      | 67.2 ± 0.3 | **68.0** ± 0.1 |
| (1, 3)        | 67.8 ± 0.4      | 67.3 ± 0.2 | **68.0** ± 0.7 |
| (2, 3)        | 67.6 ± 0.3      | **69.0** ± 0.8 | **69.0** ± 0.2 |
| (3, 1)        | 68.4 ± 0.6      | 69.1 ± 0.4 | **69.7** ± 0.6 |
| Mean accuracy | 68.1 ± 0.7      | 68.2 ± 1.0 | **68.6** ± 0.8 |

Table A.4: Accuracies (% ± sd over 3 runs) for CIFAR100 for each downsampling method with the DenseNet-BC architecture. First column represents strides at each transition block, $(2, 2)$ being the configuration of (Huang et al., 2017).

**EfficientNet experiments on CIFAR**   We evaluate DiffStride in an EfficientNet-B0 architecture (Tan & Le, 2019), a lightweight model discovered by architecture search. This architecture has seven strided convolutions. Unlike Tan & Le (2019), we do not pre-train on ImageNet, but rather train from scratch on CIFAR, which explains the lower accuracy of the baseline. As the model has seven downsampling layers, we rescale the images from $32 \times 32$ to $128 \times 128$, and only sample strides in $[1; 2]$. We run a similar experiment as in 3.2 with random strides on the two CIFAR datasets. Consistently with the results obtained with a ResNet-18, spectral pooling is much more robust to poor strides than strided convolutions, with DiffStride outperforming all alternatives.

| Init. Strides | Average Pooling | Spectral | DiffStride |
|---|---|---|---|
| $(1, 2, 2, 2, 1, 2, 1)$ | $87.2 \pm 0.1$ | $90.4 \pm 0.2$ | $\mathbf{91.1} \pm 0.0$ |
| $(1, 1, 2, 2, 2, 1, 1)$ | $89.7 \pm 0.1$ | $\mathbf{90.9} \pm 0.3$ | $90.9 \pm 0.1$ |
| $(1, 2, 2, 2, 2, 2, 1)$ | $83.7 \pm 0.2$ | $90.0 \pm 1.0$ | $90.8 \pm 0.1$ |
| $(2, 1, 2, 1, 2, 1, 1)$ | $89.2 \pm 0.2$ | $90.4 \pm 0.4$ | $\mathbf{91.1} \pm 0.1$ |
| Mean accuracy | $87.5 \pm 2.5$ | $90.4 \pm 0.6$ | $\mathbf{90.9} \pm 0.1$ |

Table A.5: Accuracies ($\% \pm$ sd over 3 runs) for CIFAR10 for each downsampling method with the EfficientNet-B0 architecture. First column represents strides at each strided convolution, with $(1, 2, 2, 2, 1, 2, 1)$ being the configuration of (Tan & Le, 2019).

| Init. Strides | Average Pooling | Spectral | DiffStride |
|---|---|---|---|
| $(1, 2, 2, 2, 1, 2, 1)$ | $55.2 \pm 0.3$ | $66.0 \pm 0.6$ | $\mathbf{66.6} \pm 0.5$ |
| $(1, 1, 2, 2, 2, 1, 1)$ | $62.0 \pm 0.7$ | $66.4 \pm 0.6$ | $\mathbf{66.6} \pm 0.3$ |
| $(1, 2, 2, 2, 2, 2, 1)$ | $46.8 \pm 2.0$ | $65.9 \pm 0.5$ | $\mathbf{66.3} \pm 0.7$ |
| $(2, 1, 2, 1, 2, 1, 1)$ | $60.4 \pm 0.1$ | $65.5 \pm 0.1$ | $\mathbf{67.0} \pm 0.1$ |
| Mean accuracy | $56.1 \pm 6.3$ | $65.9 \pm 0.5$ | $\mathbf{66.6} \pm 0.5$ |

Table A.6: Accuracies ($\% \pm$ sd over 3 runs) for CIFAR100 for each downsampling method with the EfficientNet-B0 architecture. First column represents strides at each strided convolution, with $(1, 2, 2, 2, 1, 2, 1)$ being the configuration of (Tan & Le, 2019).

**Ablation study: learning a per-dimension stride or a shared one**   We perform multi-task audio classification with either learning a single stride value for each DiffStride layer, or a different one for the time and frequency axes. The overall performance across tasks is improved when learning a different stride value for each dimension (See Table A.7).

| Setting | Multi-task | |
|---|---|---|
| Task | Shared stride | Different strides |
| Acoustic scenes | $97.4 \pm 0.3$ | $\mathbf{97.7} \pm 0.3$ |
| Birdsong detection | $\mathbf{79.7} \pm 1.0$ | $78.6 \pm 0.5$ |
| Music (instrument) | $70.7 \pm 0.5$ | $\mathbf{73.0} \pm 0.8$ |
| Music (pitch) | $86.7 \pm 0.5$ | $\mathbf{89.9} \pm 0.3$ |
| Speech commands | $\mathbf{86.8} \pm 0.1$ | $86.2 \pm 0.8$ |
| Mean Accuracy | $84.3 \pm 9.2$ | $\mathbf{85.0} \pm 8.9$ |

Table A.7: Test accuracy ($\% \pm$ sd over 3 runs) for audio classification in the multi-task (one model for all tasks) with DiffStride, when learning a single stride value (Shared stride) per layer, or a different one for each dimension (Different strides).

