# OpenReview forum: "Learning Strides in Convolutional Neural Networks"
_ICLR.cc/2022/Conference — ICLR 2022 Oral_

### Official Review · Reviewer_ubnz · 2021-10-22

**Correctness:** 3
**Technical Novelty And Significance:** 3
**Empirical Novelty And Significance:** 3
**Recommendation:** 8
**Confidence:** 4

**Main Review:**

+ The paper is well-written and the proposed formulation is sound.

+ Also extensive experiments are provided on several datasets to demonstrate the benefits of the proposed method.


- Authors demonstrate that the proposed method can recover from different initial strides and learn the optimal one. However, my most important concern is the lack of appropriate comparisons with neural architecture search approaches, since - in my view - these are the most direct competitors of the proposed method. Therefore, I would expect experiments and appropriate discussions between the cost and benefits between the proposed method and neural architecture search approaches.

- The proposed method seems to target only residual architectures. Discussion on how the proposed method could be used with other architectures would be beneficial

- Also, providing experiments with different architectures (apart from resnet) on the same dataset would also further improve the confidence on the obtained results.

- What is the actual overhead of the using the proposed method in real applications? How much does the training time and memory usage increases?

**Summary Of The Paper:**

In this work the authors introduce a differentiable stride formulation, which allows for learning the stride value. To this end, they propose learning the size of a cropping mask in the Fourier domain, which allows for learning how to perform resize in a differentiable way. Authors present experiments on several datasets on different domains (image, audio), including large scale datasets.

**Summary Of The Review:**

Overall, I think this is a good paper with an interesting approach on differential stride learning. There is no theoretical discussion and I would like to see some additional experiments, however I feel that this work is slightly above acceptance threshold, mainly given the novelty of the application.

---

> ### Author Response · Authors · 2021-11-18
> **Response to Reviewer ubnz**
>
> We thank the reviewer for their feedback.
>
> >“Authors demonstrate that the proposed method can recover from different initial strides and learn the optimal one. However, my most important concern is the lack of appropriate comparisons with neural architecture search approaches, since - in my view - these are the most direct competitors of the proposed method. Therefore, I would expect experiments and appropriate discussions between the cost and benefits between the proposed method and neural architecture search approaches.”
>
> Mentioning neural architecture search is indeed relevant. NAS and DiffStride are however complementary: NAS does not alleviate the combinatorial explosion of hyperparameters but is efficient at prioritizing exploration and finding dependencies between parameters. NAS would benefit from differentiable hyperparameters (it is stated as an explicit goal in [5]). Traditionally, NAS has not focused on extensive stride parameter search: [1] mentions the explosion of stride parameters with depth and only evaluates 3 possible strides, 1-2-3, that deteriorates the overall performance. Other neural architecture search approaches do not even cross-validate for strides due to the exponential search space of strides [2,3,4]. The goal of DiffStride is to remove this issue altogether by learning strides by backpropagation directly. The comparison of NAS without DiffStrides, NAS with DiffStrides, DiffStrides without NAS is experimentally challenging and beyond the scope of this initial paper.
>
> >“The proposed method seems to target only residual architectures. Discussion on how the proposed method could be used with other architectures would be beneficial. Also, providing experiments with different architectures (apart from resnet) on the same dataset would also further improve the confidence on the obtained results.”
>
> We focused our image experiments on ResNets as 1) it is the most widely used convolutional architecture, 2) learning strides with residual connections is not as trivial as for purely feed-forward architectures. However, we kindly bring to the reviewer’s attention that our audio classification architecture does not involve any residual connection and alternates convolutions along the time and frequency axes. Still, to address the reviewer’s concern, we added new experiments with EfficientNet-B0 in Appendix, which has seven layers of strided convolutions, and where DiffStride improves over spectral pooling and strided convolutions, consistently with our ResNet-18 results.
>
> > “What is the actual overhead of the using the proposed method in real applications? How much does the training time and memory usage increases?”
>
> We thank the reviewer for this suggestion and agree that our discussion of limitations could be improved. To this end, we added in the appendix the peak memory usage and per-step time (forward+backward) on a V100 GPU (See Table A.2 in the Appendix). This shows that while DiffStride is more costly in training due to the necessity of computing and storing gradients with respect to strides, at inference the cost is identical to that of spectral pooling.  We also extended our “Limitations'' paragraph to propose other avenues for reducing the computational load of DiffStride. We also concede that efficiency was not our primary objective in this work. Efficiency gains could be obtained by exploring compiler options and implementation choices (esp. rematerialization options for backward efficiency and separation of the real/imaginary part after FFT to benefit from optimized loss precision primitives in further processing).
>
> [1] Barret Zoph and Quoc V. Le. Neural architecture search with reinforcement learning. 2017.
>
> [2] Efficient Neural Architecture Search via Parameters Sharing. Hieu Pham, Melody Guan, Barret Zoph, Quoc Le, Jeff Dean Proceedings of the 35th International Conference on Machine Learning,
>
> [3] Liu, Hanxiao, Karen Simonyan, and Yiming Yang. "DARTS: Differentiable Architecture Search." International Conference on Learning Representations. 2018.
>
> [4] Wu, Bichen, et al. "Fbnet: Hardware-aware efficient convnet design via differentiable neural architecture search." Proceedings of the IEEE/CVF Conference on Computer Vision and Pattern Recognition. 2019.
>
> [5] Neural Architecture Optimization, Renqian Luo, Fei Tian, Tao Qin, Enhong Chen, Tie-Yan Liu (NeurIPS 2018)

---

> > ### Comment · Reviewer_ubnz · 2021-12-01
> > **Reviewer response**
> >
> > I thank the authors for providing their feedback, addressing my concerns. I still hold the opinion that this paper should be accepted.

---

### Official Review · Reviewer_34rN · 2021-11-02

**Correctness:** 4
**Technical Novelty And Significance:** 4
**Empirical Novelty And Significance:** 4
**Recommendation:** 8
**Confidence:** 4

**Main Review:**

Positives
---------

Overall I feel that the paper presents a really neat idea well. Besides a few minor issues (see below) the paper was enjoyable to read and describes the ideas well. There is thorough experimentation on a range of tasks and model architectures that demonstrate the power of the proposed approach.


Concerns
--------

Really, the only major concern I have is around the rather limited discussion of the limitations. In particular, I'd at least like some discussion (or thoughts) on why it doesn't improve densenet performance, and I'd like to see experimental results presented on the impact of the use of DiffStride during training and inference, both in terms of measurable memory usage and impact on training & inference wall time. Obviously such results can be caveated with a statement that the implementation could be improved as per the existing discussion in the limitations section.


Minor points
------------

Please check and fix algorithm 1 - I believe line 4 should be using the output from the filtering with the mask, rather than the raw FFT result. It would also be better not to reuse the $\tilde{y}$ symbol on lines 4 and 5, and it would also be helpful if the symbols (the variants of the intermediate computations $y$, $\tilde{y}$}) were clearly labelled on Fig 1.


**Summary Of The Paper:**

This paper proposes an approach to learning optimal striding parameters in convolutional networks. The proposed approach, DiffStride, is a downsampling layer that builds on spectral pooling to allow for integer output dimensions, but arbitrary strides by cropping in the Fourier domain. Unlike spectral pooling, DiffStride relaxes the parameters of the cropping mask to be differentiable, and using the stop-gradient operator in the cropping. Results show that the proposed approach can work well as a drop-in replacement for normally fixed pooling layers. It is also demonstrated that from different random initialisations a variety of different pooling approaches can be learned that acheive similar performance. Use of a regularisation term the attempts to encourage time and space efficiency reduces this variability and allows accuracy to be traded off.

**Summary Of The Review:**

Overall, as can be seen from my review I'm satisfied that this paper makes a good contribution.

---

> ### Author Response · Authors · 2021-11-18
> **Response to Reviewer 34rN**
>
> We thank the reviewer for their feedback.
>
> >“In particular, I'd at least like some discussion (or thoughts) on why it doesn't improve densenet performance, “
>
> We do not have a definitive answer for the lack of improvement for spectral DenseNet. Clearly, the limited number of downsampling layers (only 2) naturally reduces the space of stride configurations to a few, equivalent ones when sampling strides in [1; 3]. We added a mention of this in the “Limitations” paragraph. In future work, we aim to explore whether DenseNet/Resnet hybrid networks like DSNet [1] could explain whether the location of the striding layers or the dense shortcut modules explain the lack of improvement from spectral pooling. We added new experiments with EfficientNet-B0 in Appendix, which has seven layers of strided convolutions, and where DiffStride improves over spectral pooling and strided convolutions, consistently with our ResNet-18 results.
>
> > “and I'd like to see experimental results presented on the impact of the use of DiffStride during training and inference, both in terms of measurable memory usage and impact on training & inference wall time. Obviously such results can be caveated with a statement that the implementation could be improved as per the existing discussion in the limitations section. “
>
> We agree that our discussion of limitations could be improved. To this end, we added in the appendix the peak memory usage and per-step time (forward+backward) on a V100 GPU (See Table A.2 in the Appendix). This shows that while DiffStride is more costly in training due to the necessity of computing and storing gradients with respect to strides, at inference the cost is identical to that of spectral pooling.  We also extended our “Limitations” paragraph to propose other avenues for reducing the computational load of DiffStride. In particular, a main computational bottleneck is the need to operate in the Fourier domain. We mention a potential alternative, which has already been employed for non-learnable Spectral Pooling, namely the use of the Hartley transform, a real-valued counterpart to the Fourier transform. We also concede that efficiency was not our primary objective in this work. Efficiency gains could be obtained by exploring compiler options and implementation choices (esp. rematerialization options for backward efficiency and separation of the real/imaginary part after FFT to benefit from optimized loss precision primitives in further processing).
>
> > “Please check and fix algorithm 1 - I believe line 4 should be using the output from the filtering with the mask, rather than the raw FFT result. It would also be better not to reuse the  symbol on lines 4 and 5, and it would also be helpful if the symbols (the variants of the intermediate computations , }) were clearly labelled on Fig 1.”
>
> We thank the reviewer for pointing out this typo. We fixed Algo 1 and clarified the notations. We also modified Figure 1 accordingly.
>
> [1] Zhang, C., Benz, P., Argaw, D.M., Lee, S., Kim, J., Rameau, F., Bazin, J.C. and Kweon, I.S., 2021. Resnet or densenet? introducing dense shortcuts to resnet. In Proceedings of the IEEE/CVF Winter Conference on Applications of Computer Vision (pp. 3550-3559).

---

### Official Review · Reviewer_e1mG · 2021-11-02

**Correctness:** 4
**Technical Novelty And Significance:** 3
**Empirical Novelty And Significance:** 4
**Recommendation:** 8
**Confidence:** 3

**Main Review:**

PROs
- The paper is very well written and pretty much clear at a first read. All the introduced components are properly motivated intuitively, and figures and algorithms really help the reader in understanding.
- The authors excel in framing their work within related art. Specifically, they discuss both standard alternatives such as pooling, as well as alternatives based on fractional stride. For all of them, issues are properly pointed out such that the motivation behind this paper is very clear.
- The paper is, to the best of my effort, technically sound. In terms of novelty, one might argue it being a bit incremental over SpectralPool, as it substitutes the cropping hyperparameters with learnable values. However, i) this step is not trivial, as the cropping operation is non-differentiable, and the authors introduce a solution to that and ii) this modification enables a significant improvement in performances, registered across a number of datasets.
- The authors support their proposal by carrying out experiments on multiple datasets. Specifically, 6 datasets are employed for audio recognition and 3 (CIFAR-10, CIFAR-100 and Imagenet) are used for image recognition. In my opinion, for a paper of this type, attacking a very fundamental and obiquitous operator in modern architectures, being able to showcase improvements on non-toy datasets such as Imagenet is remarkable and noteworthy.
- For every experiment, the authors report the mean and standard deviation over multiple trials, strenghtening the reliability of the conclusions.
- The authors properly discuss limitations of their work, by highlighting the computational cost, some failure cases on DenseNets and implementation challenges on specific types of hardware.

---
CONs

I think this paper is very solid, and I consider the following points as minor concerns.
- In Sec 3.1, the authors claim that as "DiffStride learns different strides for the time and frequency axes", and as such "this justifies using a different parameter for each dimension rather then sharing strides". However, this validation is flawed. Just because the model takes advantage of this flexibility, it doesn't necessarily mean that it is beneficial in the end. This might be the case if we completely trusted gradient descent to reach global optimums of the cost function, which is not the case. Indeed, the final stride configuration might depend much on the initialization, as also suggested by the experiment in Fig. 3. Therefore, to validate such a claim the authors should simply include a baseline model where DiffStride is applied with shared striding parameters for the time and frequency axes.
- Fig. 4 would be much clearer if on the x axis MACs or FLOPs were represented, instead of the regularization term. This would help quantify how much computation DiffStride can save without losing significant performances. As it is, the value of the regularization term dos not tell much about actual computational cost.
- To my understanding, within experiments the authors substitute a few downsampling layers (not even all of them) in a network with DiffStride. As a reader, I would have expected that every convolutional layer would be equipped with its own learnable striding. I grasp that this may not be practical but I don't have a clear view of the reason. The authors should consider adding some motivation behind this choice.
- Another point that may be worth discussing as an extension/future work. May the DiffStride technique be employed within a conditional computation framework, by predicting the striding parameters conditioned on the current example? I think that this strategy might succeed especially when aiming at optimizing the computational cost of a model. I do not expect such an extention to be implemented and tested within the rebuttal period. I would just like the authors' opinion about that (potential, concerns etc).
- In Tab. 1, the Speaker Id dataset provides no information whatsoever, as all methods score a perfect accuracy of $100.0 \pm 0.0$. I suggest the authors to remove this dataset, as it is apparently beyond trivial, and including it downgrades the quality of the experiment rather than increasing it.
- On CIFAR-100, the authors show that SpectralPool is significantly better than strided convs even with the same downsampling hyperparameters. Can the authors provide an intuition for this behavior?

**Summary Of The Paper:**

In this paper, the authors propose DiffStride, a technique for learning the stride of downsampling operations in neural networks by gradient descent. Specifically, similarily to SpectralPool, the feature map is transformed to the frequency domain by a Discrete Fourier Transform, where it is then cropped according to learnable parameters. The authors envision a simple soft-relaxation of the cropping in order to allow the gradient to flow towards cropping parameters. Performances against both standard and random stride policies are reported on a number of datasets, both for audio and image recognition. Moreover, the authors introduce a regularization objective that penalizes the use of small strides, in the interest of encouraging downsampling for improving computational and memory cost.

**Summary Of The Review:**

Overall, I recommend the paper for acceptance. The contribution is novel and clearly motivated, and experimental results are encouraging across different datasets. I think this paper can be of interest for many within the ICLR community. There are some improvement points as discussed above, but in my opinion the strengths clearly outnumber the weaknesses.

---

> ### Author Response · Authors · 2021-11-18
> **Response to Reviewer e1mG**
>
> We thank the reviewer for their feedback.
>
>
> > “In Sec 3.1, the authors claim that as "DiffStride learns different strides for the time and frequency axes", and as such "this justifies using a different parameter for each dimension rather then sharing strides". However, this validation is flawed. Just because the model takes advantage of this flexibility, it doesn't necessarily mean that it is beneficial in the end. This might be the case if we completely trusted gradient descent to reach global optimums of the cost function, which is not the case. Indeed, the final stride configuration might depend much on the initialization, as also suggested by the experiment in Fig. 3. Therefore, to validate such a claim the authors should simply include a baseline model where DiffStride is applied with shared striding parameters for the time and frequency axes.”
>
> We thank the reviewer for pointing out this claim, and we agree that the existence of equivalence classes as shown in Fig. 3 invalidates it. To address this concern, we also trained the multi-task audio classification model with a shared stride for both axes and report the results in a new Table A.7. This shows that sharing the strides reduces the average performance on this task, and now refer to this result in the text to support our choice of not sharing the strides.
>
> >“”Fig. 4 would be much clearer if on the x axis MACs or FLOPs were represented, instead of the regularization term. This would help quantify how much computation DiffStride can save without losing significant performances. As it is, the value of the regularization term dos not tell much about actual computational cost.“”
>
>
> We agree that MACs or FLOPs would be more interpretable. In the limited time of the rebuttal period, this appeared challenging as Tensorflow profiling tools do not track complex-valued operations (and thus do not count DFT and IDFT). Moreover, we wanted to disentangle the “theoretical” computational cost that depends only on strides (as illustrated by the regularization term) from implementation choices. However, we agree that Fig.4 does not provide enough hints on the real, practical performance of DiffStride in terms of wall time and memory usage. To address this issue, we added in the appendix the peak memory usage and per-step time (forward+backward) on a V100 GPU (see new Table A.2 in the Appendix). This shows that while DiffStride is more costly in training due to the necessity of computing and storing gradients with respect to strides, at inference the cost is identical to that of spectral pooling. We also extended our “Limitations” paragraph to propose other avenues for reducing the computational load of DiffStride. We also concede that efficiency was not our primary objective in this work. Efficiency gains could be obtained by exploring compiler options and implementation choices (esp. rematerialization options for backward efficiency and separation of the real/imaginary part after FFT to benefit from optimized loss precision primitives in further processing).
>
> >“To my understanding, within experiments the authors substitute a few downsampling layers (not even all of them) in a network with DiffStride. As a reader, I would have expected that every convolutional layer would be equipped with its own learnable striding. I grasp that this may not be practical but I don't have a clear view of the reason. The authors should consider adding some motivation behind this choice.”
>
> Our audio classification architecture performs downsampling at every convolution layer, so we do the same with DiffStride. ResNets first downsample with a max-pooling operator with stride 2 and then only have a few strided convolutions (namely, 4), regardless of their depth (i.e. switching from a ResNet18 to a ResNet50 only adds non-strided convolutions between downsampling layers), and we replace all of them on ImageNet, while only replacing 3 out of 4 on CIFAR. This is necessary to process small inputs, such as 32x32 CIFAR images, for which only a few downsampling layers are enough to reach a spatial dimension of 1x1. We do not replace the first max-pooling operator to keep our comparison consistent (we are only replacing strided convolutions in this work).
> Currently, we only replace strided convolutions with DiffStride modules. In future work, we would like to replace non-strided convolutions DiffStride layers initialized with a stride of 1. We however will be in position to perform these changes only after reducing the cost of DiffStride (see new appendix A.2) by reducing the compute overhead due to non optimized complex operations (e.g. with Hartley transform, a real-valued counterpart to the Fourier transform) and the memory overhead due to dual spatial/spectral representation (e.g. with spectral non-linearities, exploration of rematerialization options).
>
> (Cont. in next comment)

---

> > ### Author Response · Authors · 2021-11-18
> > **Response to Reviewer e1mG (cont.)**
> >
> > >“Another point that may be worth discussing as an extension/future work. May the DiffStride technique be employed within a conditional computation framework, by predicting the striding parameters conditioned on the current example? I think that this strategy might succeed especially when aiming at optimizing the computational cost of a model. I do not expect such an extention to be implemented and tested within the rebuttal period. I would just like the authors' opinion about that (potential, concerns etc).”
> >
> >
> > This is indeed a promising direction that is unlocked by casting strides as learnable parameters of the model. We have considered this during the late stages of development of DiffStride, however this comes with several challenges:
> >
> > **Benefits**: For instance this is an important argument to have adaptive striding for the speech inputs. There is speed variability among speakers and psychoacoustical studies that demonstrate fast online adaptation of humans [1]. Image classification would also benefit from being able to adaptively select the downsampling factors, to provide e.g. invariance to scale or to reduce the computational cost for tasks where low frequencies (e.g. textures) are enough to discriminate between images which allows for aggressive downsampling.
> >
> > **Challenges**: The first challenge is computational. Even though DiffStride changes the shapes of intermediate shapes between each batch, they are constant between backward passes which allows for batching. On the other hand, predicting strides on a per-example basis would make batching difficult, and would require padding all intermediate representations to the largest dimension in the batch. The second challenge is in terms of modelling: what kind of architecture would be appropriate to predict such strides? It would ideally be a lightweight model that can aggregate statistics over the whole input, such as a small CNN. Finally, in the main model, it is unclear whether the same convolutional kernels could be simultaneously applied to examples with different striding, i.e. inputs with different truncated frequencies.
> > We thank the reviewer for mentioning this possibility and added a mention in our conclusion.
> >
> >
> > > “In Tab. 1, the Speaker Id dataset provides no information whatsoever, as all methods score a perfect accuracy of 100.0±0.0  I suggest the authors to remove this dataset, as it is apparently beyond trivial, and including it downgrades the quality of the experiment rather than increasing it.”
> >
> > We agree that these results do not bring anything to the paper. We kept it at first for completeness, but removed the mention of this dataset in our revised manuscript.
> >
> > > “On CIFAR-100, the authors show that SpectralPool is significantly better than strided convs even with the same downsampling hyperparameters. Can the authors provide an intuition for this behavior?”
> >
> > Spectral pooling and DiffStride act as lowpass filters before downsampling and remove the high frequency components of inputs and preserve the low frequencies. This is unlike strided convolutions that introduce aliasing during the downsampling step as was demonstrated in [2,4]. In [2], Zhang they also shows that performing lowpass filtering before downsampling improves the robustness of classical, spatial-pooled convolutional neural networks. Besides, signals in several modalities tend to be biased towards low frequencies as we mentioned in Section 2.3. This is supported by image classification experiments in [3] where spectral pooling outperforms alternatives, which the authors attribute to preserving low-frequency information. Our results with spectral pooling confirm these results on all tasks and architectures that we consider [3]. Moreover, the observation made in Paragraph “Regularizing the complexity” that rounding fractional strides learned by DiffStride and using them to train a model with strided convolutions gives very poor performance furthermore corroborates the higher robustness to strided configurations of pooling the frequency domain.  We now mention aliasing in introduction and Section 2.3.
> >
> > [1] On-line plasticity in spoken sentence comprehension: Adapting to time-compressed speech. Patti Adank and Joseph T.Devlin. NeuroImage 2010.
> >
> > [2] Making convolutional networks shift-invariant again. In ICML, 2019 Richard Zhang
> >
> > [3] Spectral representations for convolutional neural networks. In Neurips, 2015 O Rippel, J Snoek, RP Adams
> >
> > [4] How Convolutional Neural Networks Deal with Aliasing, Ribeiro and Schoen, ICASSP'21.

---

> > > ### Comment · Reviewer_e1mG · 2021-11-30
> > > **Post rebuttal comment**
> > >
> > > I was already convinced of the quality of the paper.
> > > Reading other reviews and comments by the authors did nothing other than reinforcing my initial assessment. The authors
> > >
> > > - ablated the importance of disentangling strides among the two axis, with positive results;
> > > - added an analysis of computational and memory cost of their model, and mention it within a "limitations" section;
> > > - clarified my doubts about why they limit (in resnets) the employment of DiffStride to several layers only;
> > > - provided interesting insights about the possibiliy to condition the strides on the input itself;
> > > - explain why spectral pool outperforms aliased strided convs given equality in stride.
> > >
> > > In summary, I strongly advocate for accepting this paper.

---

### Official Review · Reviewer_BTMH · 2021-11-03

**Correctness:** 4
**Technical Novelty And Significance:** 3
**Empirical Novelty And Significance:** 3
**Recommendation:** 8
**Confidence:** 4

**Main Review:**

The idea is interesting and makes sense to me. The writing is clear and easy to understand.  However, from the experiment results (especially table3 and 4), it seems that the proposed method has marginal improvement compared to regular strided conv and spectral pooling baselines. It seems that the default setting can already achieve very good results on ImageNet and how it will affect the model performance when the model is large enough remains unclear. The benefit of learnable strides is not fully demonstrated. It would be great if the authors could implement one or more future works in Sec. 4 to showcase its capability further and show the proposed module's overheadein detail to let the readers better understand its limitation.



**Summary Of The Paper:**

This paper proposes DiffStride as a drop-in replacement to standard downsampling layers. It extends previous work Spectral Poolnig and learns the size of the cropping box in the frequency domain by backpropagation. Experiments are conducted on audio and image classification, and the results show that the model can learn non-integer stride and adapt different initial stride well.

**Summary Of The Review:**

Overall I think this paper presents a novel idea to learn stride in the downsampling layer. However, the current results are not good enough to showcase its effectiveness by making the stride learnable. The authors may want to find more ways, as discussed in Sec.4, to show its value.

-- Post rebuttal
After reading the authors' response, I raise my rating to 8.

---

> ### Author Response · Authors · 2021-11-18
> **Response to Reviewer BTMH**
>
> We thank the reviewer for their feedback.
>
> >“However, from the experiment results (especially table3 and 4), it seems that the proposed method has marginal improvement compared to regular strided conv and spectral pooling baselines. It seems that the default setting can already achieve very good results on ImageNet and how it will affect the model performance when the model is large enough remains unclear. The benefit of learnable strides is not fully demonstrated. ”
>
> We acknowledge that for image classification, DiffStride does not improve significantly over the best stride configurations found in the literature by cross-validation. However, this is not the main goal of our contribution. Our contribution is the first downsampling layer with a learnable stride, which replaces multiple training runs for cross validation with a single training run with stride optimization. This is a clear benefit we stress in the introduction " even when initializing strides randomly, our model converges to the best performance obtained with the properly cross-validated strides of He et al".
>
> The hypothesis that we test in our image classification experiments is the following: “can learning strides by backpropagation remove the need for cross-validation?”. The fact that our ResNet-18 architecture becomes insensitive to the choice of strides when using DiffStride, while it is significantly more affected by poor choices of strides for baselines, supports this claim, with a variance over initializations an order of magnitude below that of strided convolutions. We added **new experiments with EfficientNet-B0 in Appendix**, that confirm our findings.
>
> The fact that DiffStride does not discover a significantly better stride configuration for ResNet18 than the configuration of He et al. could be underwhelming or, more likely, an indication of the extensive exploration of hyperparameters by the ResNet authors. This is unlike the less explored classifier we use for single and multi-task audio classification, where DiffStride outperforms strided convolutions on four tasks out of five.
>
> What we hope will be the impact of our contribution on the community is that designing new convolutional architectures (or other architectures that involve downsampling), where finding the best strides configuration would require tedious cross-validation, will be easier with DiffStride which can retrieve the optimal performance in a single training without cross-validation.
>
> >“It would be great if the authors could implement one or more future works in Sec. 4 to showcase its capability further and show the proposed module's overheadein detail to let the readers better understand its limitation”
>
> Extending DiffStride to 1D or 3D requires extending our approach to new modalities, baselines and datasets, which we leave for future work.
>
> However, we agree that our discussion of limitations could be improved. To this end, we added in the appendix the peak memory usage and per-step time (forward+backward) on a V100 GPU (See Table A.2 in the Appendix). This shows that while DiffStride is more costly in training due to the necessity of computing and storing gradients with respect to strides, at inference the memory cost and run time are identical to those of spectral pooling.  We also extended our “Limitations” paragraph to propose other avenues for reducing the computational load of DiffStride. We also concede that efficiency was not our primary objective in this work. Efficiency gains could be obtained by exploring compiler options and implementation choices (esp. rematerialization options for backward efficiency and separation of the real/imaginary part after FFT to benefit from optimized loss precision primitives in further processing).
>
> >“Overall I think this paper presents a novel idea to learn stride in the downsampling layer. However, the current results are not good enough to showcase its effectiveness by making the stride learnable.”
>
> DiffStride is a first attempt at learning strides by backpropagation, which had not been shown to be doable in the past. We experiment on 8 classification tasks both on audio and images, with different architectures, and show the following:
> * DiffStride removes the need for cross-validating strides, and allows recovering the performance of the best configuration found by cross-validation in previous work on ResNets (likely the most widely studied convolutional architecture), in a single training phase, regardless of the initialization.
> * For less explored architectures (see our audio classification experiments), DiffStride finds new stride configurations that outperform strided convolutions significantly.
> * DiffStride allows to  regularize computation and memory cost of the models directly by backpropagation.
> We additionally showcase the interpretability capabilities in Table 2, where the strides can provide insights about frequency biases in datasets and tasks and singling out the relevant scales used by a model.

---

### Author Response · Authors · 2021-11-18
**Response to reviewers**

We thank the reviewers for their feedback and suggestions. This helped improve our submission and better support our claims. We summarize here the main changes brought to the manuscript:

* Addressing a concern of all reviewers, we added a comparison between DiffStride and baselines in terms of wall time and memory usage, and extended the description of limitations.
* Following the suggestions of Reviewer e1mG, we removed the LibriSpeech dataset from audio classification experiments.
* To better support learning independent stride values for each dimension, we added an ablation study that compares sharing strides vs learning independent ones on audio classification.
* We added results with EfficientNet-B0 in Appendix to address a concern of Reviewer ubnz.

We did our best to address the main concerns raised by reviewers and we hope that these improvements will be taken into consideration. To better highlight the changes brought to our manuscript, they appear in *brown* in the revised manuscript. Under acceptance, we will revert this change in color. We also answer each reviewer separately below.

---

### Decision · Program_Chairs · 2022-01-20

**Decision:**

Accept (Oral)

**Comment:**

The paper proposes a method to learn the stride of downsampling in deep networks using a  gradient-based learning approach. The main idea is to work in the frequency domain and to learn the cropping mask in that domain. The authors also introduce a regularization for applications seeking computationally and memory efficiency. The authors investigate the interest of the approach on a number of datasets with audio and image data.

The reviewers praised the paper, appreciating the elegance of the approach and the effort made to thoroughly evaluate it. The reviewers also appreciated the clarity of the exposition and the care in the reporting of the results. The reviewers also expressed some concerns about several choices of the design (stride sharing) and the lack of detail in some experiments (computational /memory efficiency). Finally, the reviewers wished the paper had more theoretical grounding.

The authors submitted detailed responses to the reviewers' comments. After reading the responses, updating the reviews, and discussion, the reviewers found that the responses were ‘reinforcing [their] initial assessments' and their several concerns were satisfactorily addressed. Moreover several of the reviewer’s suggestions clearly already led to an improved manuscript with very thorough experimental evaluation and simpler approaches for stride sharing.

The paper proposed an elegant, learning-based, approach to one of the most important design choices in deep network architecture design: the strides in the convolutions. The authors provided a careful and thorough experimental evaluation, and moreover improved it during the review process following the reviewer’s feedback.

Accept, definitely.